# Multi-omic identification of perineurial hyperplasia and lipid-associated nerve macrophages in human polyneuropathies

Michael Heming [1,13], Anna-Lena Börsch [1,13], Jolien Wolbert [1,13], Christian Thomas [2], Anne K. Mausberg [3], Fabian Szepanowski [3], Bianca Eggert [3], I-Na Lu [1], Julia Tietz [1], Finja Dienhart [1], Maja Meschnark [1], Jan-Kolja Strecker [4], Michael Glatza [2], Carolina Thomas [5], Noemi Gmahl [1], Christine Dambietz [1], Michael Müther [6], Anne-Kathrin Uerschels [7], Kathy Keyvani [8], Jens Minnerup [4], Kathrin Doppler [9], Nurcan Üçeyler [9], Julieta Aprea [10], Andreas Dahl [10], Ruth Stassart [5], Robert Fledrich [11], Heinz Wiendl [12], Claudia Sommer [9], Mark Stettner [3] & Gerd Meyer zu Hörste [1] ✉

Diseases affecting multiple peripheral nerves, termed polyneuropathies (PNPs), are common, mechanistically heterogeneous, and their causes are challenging to identify. Here, we integrated single-nucleus transcriptomics of peripheral nerves from 33 human PNP patients and four controls (365,708 nuclei) with subcellular spatial transcriptomics. We identified nerve cell type markers and uncovered unexpected heterogeneity of perineurial cells. PNPs shared a loss of myelinating and an increase in repair Schwann cells and endoneurial lipid-phagocytizing macrophages. Transcriptional changes affected multiple cells outside of the endoneurium across PNPs, suggesting PNPs as 'pan-nerve diseases'. Spatially, PNPs—particularly those mediated by autoimmunity—exhibited focal perineurial hyperplasia and increased expression of *CXCL14*, identified as perineurial cell marker. Multi-omic characterization of human nerve biopsies thus identified novel mechanisms in PNPs with diagnostic potential.

Polyneuropathies (PNPs) denote diseases of multiple peripheral nerves and range among the most common neurological diseases[1]. Patients affected by PNPs often suffer from progressive sensory and motor impairment with high socio-economic impact[2,3]. The underlying causes of PNPs are very diverse[4–6]. Despite a comprehensive diagnostic work-up, the etiology of PNPs often remains unclear (up to 20–30%)[5]. A biopsy of the sensory sural nerve at the lateral ankle is often the final diagnostic step, but even such a biopsy does not identify the cause of PNPs in many patients[7]. Fully exploiting this precious biomaterial for mechanistic understanding and diagnostic potential is especially important to detect treatable causes such as immune-mediated neuropathies, which account for up to 10% of the PNPs[4,5]. We and others

[1]Department of Neurology, University Hospital Münster, Münster, Germany. [2]Institute of Neuropathology, University Hospital Münster, Münster, Germany. [3]Department of Neurology, Center for Translational Neuro- and Behavioural Sciences (C-TNBS), University Hospital Essen, Essen, Germany. [4]Department of Neurology, University Hospital Schleswig-Holstein, Lübeck, Germany. [5]Paul-Flechsig-Institute of Neuropathology, University Hospital Leipzig, Leipzig, Germany. [6]Department of Neurosurgery, University Hospital Münster, Münster, Germany. [7]Department of Neurosurgery, University Hospital Essen, Essen, Germany. [8]Department of Neuropathology, University Hospital Essen, Essen, Germany. [9]Department of Neurology, University Hospital Würzburg, Würzburg, Germany. [10]DRESDEN-concept Genome Center, Center for Molecular and Cellular Bioengineering, Dresden University of Technology, Dresden, Germany. [11]Institute of Anatomy, University of Leipzig, Leipzig, Germany. [12]Clinic of Neurology and Neurophysiology, University of Freiburg, Freiburg, Germany. [13]These authors contributed equally: Michael Heming, Anna-Lena Börsch, Jolien Wolbert. ✉e-mail: gerd.meyerzuhoerste@ukmuenster.de

previously performed single-cell transcriptomic analyses of rodent peripheral nerves and thereby deepened our mechanistic understanding of peripheral nerve diseases[8–13]. However, human peripheral nerves have not been characterized in health or disease using similar technologies.

Here, we provide a single-cell atlas of 37 human peripheral nerves (365,708 nuclei) integrated with spatial transcriptomics. In addition to identifying novel cell type markers, we redefined the unexpected heterogeneous perineurial cells. In diseases, repair and damage to Schwann cells (SC) and lipid-phagocytosing macrophages increased across PNP etiologies. PNPs also affected multiple cell types beyond the endoneurium. Specifically, we identified perineurial hyperplasia, especially in immune-mediated polyneuropathies, and this was associated with increased expression of the novel perineurial marker CXCL14. Multi-omic profiling of human peripheral nerves thus identified novel mechanisms with diagnostic potential.

## Results

### Comprehensive species-specific single-nucleus transcriptional atlas of human sensory nerves

We sought to better understand human PNPs and first created a large single-nucleus transcriptomics atlas of human sural nerves of 37 donors (Fig. 1A). Sural nerves from 33 PNP patients and four controls were collected in three centers (Supplementary Fig. 1A and Supplementary Data 1). Control samples (CTRL) were residual material of surgical sural nerve autografts ('interpositions') from patients with traumatic nerve injuries. We performed single-nucleus RNA-sequencing (snRNA-seq) of all samples ("Methods", Supplementary Fig. 1B and Supplementary Data 2). After analytical removal of doublets and low-quality nuclei and batch correction, this resulted in 365,708 total high-quality nuclei (Supplementary Fig. 1C, D and Supplementary Data 2). We then clustered the nuclei, resulting in 24 individual clusters (Fig. 1B and Supplementary Fig. 1E), and annotated them using predefined marker genes (Supplementary Fig. 2A and Supplementary Data 3) and automatic annotation based on rodent data[9,10,14] (Supplementary Fig. 2B). We identified Schwann cells (S100B, SOX10) of both myelinating (mySC; MPZ, MBP, PRX) and non-myelinating (nmSC; NCAM1, L1CAM) types (Fig. 1B and Supplementary Fig. 2A). Schwann cells expressed features of repair[15] (repairSC; NGFR, ATF3, GDNF, RUNX2) and of damage (damageSC; EGR1, FOS, JUN). Moreover, we found endoneurial (endoC; SOX9, PLXDC1, ABCA9), perineurial (periC1-3; SLC2A1/GLUT1, KRT19, CLDN1), and epineurial (epiC; CCBE1, COMP) cells (Fig. 1B and Supplementary Fig. 2A). This perineurial cell heterogeneity was greater than previously described in rodents[8–10,12]. The periC3 cluster was distinguished from other perineurial clusters by genes associated with extracellular matrix and collagen fibril organization (Supplementary Fig. 2C). Vascular cells included vascular smooth muscle cells (VSMC; ACTA2, CARMN), pericytes (PC1-2; RGS5, PDGFRB), and endothelial cells (EC; EGFL7, PECAM1) (Supplementary Fig. 2A). Based on known markers (Supplementary Fig. 2A) and published data[14] (Supplementary Fig. 2B), endothelial cells (EC) separated into a venous (ven_EC: PLVAP, ACKR1) to capillary (capEC: ABCG2, MFSD2A) to arterial (artEC: SEMA3G, HEY1, GJA5) continuum and lymphatic endothelial cells (LEC; PROX1, LYVE1, FLT4). We identified a cluster of venous/capillary EC (ven_cap_EC2), which expressed the blood-nerve barrier (BNB) markers ABCB1 and SLC1A1[16], blood-brain barrier marker MFSD2A[17], and tight junction transcript GJA1 and therefore likely represented EC of the BNB (Fig. 1B and Supplementary Fig. 2A). Nerve-associated leukocytes were mainly of myeloid lineage, outnumbering T/NK cells and B cells (Fig. 1B and Supplementary Fig. 2A). We thus created a transcriptional cellular atlas of human peripheral nerves, considerably extending rodent studies[8–13].

We systematically compared cell type markers in humans with their published rodent counterparts[8–10] (Supplementary Data 4–6). As expected, the majority of transcripts were shared between species. However, several genes expressed in mySC (MLIP), nmSC (GRIK3, PRIMA1), and periC (CXCL14) were undescribed in rodent or human literature (Fig. 1C). When reanalyzing rodent sc/snRNA-seq datasets[8–10], we detected Mlip in mySC in two studies[9,10]; albeit previously unmentioned (Fig. 1C and Supplementary Data 4–6). In contrast, Grik3, Prima1, and Cxcl14 were not detectable in rodent sc/nRNA-seq datasets (Supplementary Fig. 2D and Supplementary Data 4–6). We thus identified novel transcripts across nerve-associated cells.

### Spatially confirming novel cell type markers across species

For morphological confirmation, we first employed in situ amplification-based spatial transcriptomics with subcellular resolution ('spatial-seq')[18]. We designed a custom marker panel of 99 known (e.g., PRX) and novel (e.g., GRIK3) transcripts (Supplementary Data 7) while excluding highly transcribed genes to prevent optical overcrowding ("Methods"). Visualizing these targets in a formalin-fixed paraffin-embedded (FFPE) cross-section of a human sural nerve graft (CTRL) replicated the overall cellular organization of peripheral nerves. We matched each cell in spatial-seq to an snRNA-seq cluster, aggregated these into 8 larger groups, and then visualized these groups (Fig. 1D). As expected, the nerve consisted of multiple fascicles formed by endoneurial cell types (SC, endoC) surrounded by perineurial cells (periC). Vascular cell types formed vessels of their respective morphology (Fig. 1D). Epineurial cells (epiC) were located in between fascicles and were interspersed with immune cells that were rare in the endoneurial areas (Fig. 1D). We thus morphologically confirmed the unbiased annotation of human peripheral nerve cell clusters.

In addition, previously undescribed perineurial markers (CXCL14) indeed colocalized with known perineurial cell transcripts (CLDN1, SLC2A1, KRT19)[11] in the perineurium (Fig. 1E). In addition, we confirmed CXCL14 expression in the perineurium using immunofluorescence (Fig. 1F). We found that CXCR4—the receptor for CXCL14—was primarily expressed in the B and T/NK clusters (Supplementary Fig. 3A). In addition, we performed a computational analysis of cell-cell interactions and found that perineurial cells, particularly the periC2 cluster, were predicted to interact with B cells via CXCL14-CXCR4 binding (Supplementary Fig. 3B). Immune cells may thus signal to perineurial cells in human peripheral nerves.

The BNB markers ABCB1 and SLC1A1, expressed by the ven_capEC2 cluster in snRNA-seq, were enriched in endoneurial vessels together with known EC markers (PECAM1, EGFL7)[11,19] (Fig. 1G). This supported that the ven_capEC2 cluster represents BNB cells. Other vascular cell markers (e.g., SEMA3G in artEC) were associated with their respective vessel type (Supplementary Fig. 3C). Known pan-SC markers (SOX10, S100B, EGR2)[11] were expressed across the endoneurium (Supplementary Fig. 3D). Known markers of mySC (PRX, CDH1, SLC36A2)[9,13] colocalized with MLIP identified in the mySC cluster (Fig. 1H). Markers of nmSC (CDH2, L1CAM, CHL1)[12,19,20] colocalized with novel transcripts of the nmSC cluster (PRIMA1, GRIK3) in human nerve sections (Fig. 1H).

Next, we tested whether the novelty of these markers (CXCL14, PRIMA1, GRIK3) and their orthologues was due to technical differences between published and our dataset or due to human-specific expression by combining RNA in situ hybridization with immunofluorescence (RNA-ISH/IF) in rodent material. While Grik3 was not detectable in rodent sc/snRNA-seq data (Supplementary Fig. 2D and Supplementary Data 4–6). Grik3 was expressed in murine nerves using RNA-ISH/IF, albeit at a much lower intensity than in positive control tissues (Supplementary Fig. 3E). Grik3 mostly colocalized with the nmSC marker L1CAM (Supplementary Fig. 3E), consistent with our findings in the

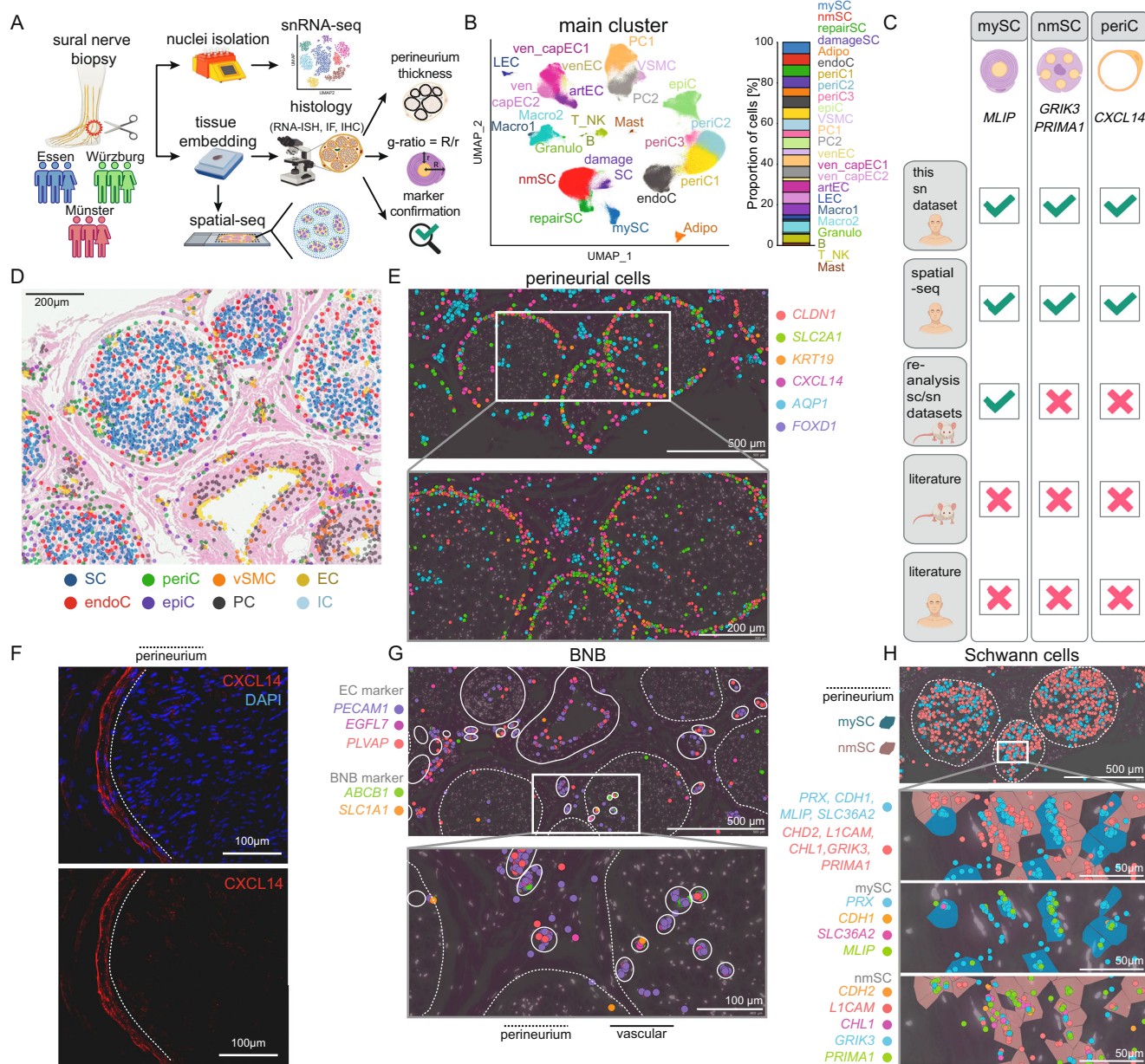

**Fig. 1 | Defining the species-specific cellular landscape of human peripheral nerves. A** Schematic overview of the experimental procedure. Sural nerve biopsies were collected from 33 polyneuropathy (PNP) patients and four controls (CTRL) from three German centers (Essen, Münster, Würzburg) and processed by single-nucleus RNA-sequencing (snRNA-seq). In addition, subcellular spatial transcriptomics (spatial-seq) was performed in a subgroup of eight patients. All 37 tissues were histologically characterized by quantifying myelin thickness (g-ratio), number of intactly myelinated axons, and axon diameters. Perineurium thickness was measured in 12 CIDP and 18 CTRL patients using immunohistochemistry (IHC) targeting epithelial membrane antigen. Markers were confirmed using RNA in situ hybridization (RNA-ISH) and immunofluorescence (IF). **B** UMAP of 365,708 high-quality nuclei from 37 human sural nerves, showing 24 main clusters. **C** Schematic depicting if marker genes of human myelinating Schwann cells (mySC), non-myelinating Schwann cells (nmSC), and perineurial cells (periC) found in this dataset using snRNA-seq and spatial-seq were detected when reanalyzing published rodent datasets (Yim et al., Gerber et al., Wolbert et al., Supplementary Data 4–6) when performing RNA-ISH/IF histology in rodents and whether these markers were previously described in rodent or human literature. **D**, **E** Spatial-seq was performed on sural nerve samples from a total of eight patients. Representation sections from CTRL patient S24 are shown. **D** Representative section of an H&E staining overlaid with spatial-seq showing predicted clusters in spatial-seq of a sural nerve cross-section. Each dot represents one cell. **E** Representative spatial-seq images of selected genes expressed in perineurial cells. **F** Immunofluorescence staining was performed on a minimum of two sural nerve sections per patient from a total of four patients. Representative immunofluorescence image showing CXCL14 staining in perineurial cells from CIAP patient S10. The dotted line marks the perineurium. **G**, **H** Spatial-seq was performed on sural nerve samples from a total of eight patients. Representation sections from CTRL patient S24 are shown. Representative spatial-seq images of selected genes expressed in (**G**) cells of the blood nerve barrier (BNB) and (**H**) in mySC (blue highlight) and nmSC (red highlight). Each dot represents the expression of one transcript, a dotted line marks the perineurium, and a solid line surrounds individual vessels. Icons were created in BioRender. Meyer zu Hörste, G. (2025) https://BioRender.com/l74v346.

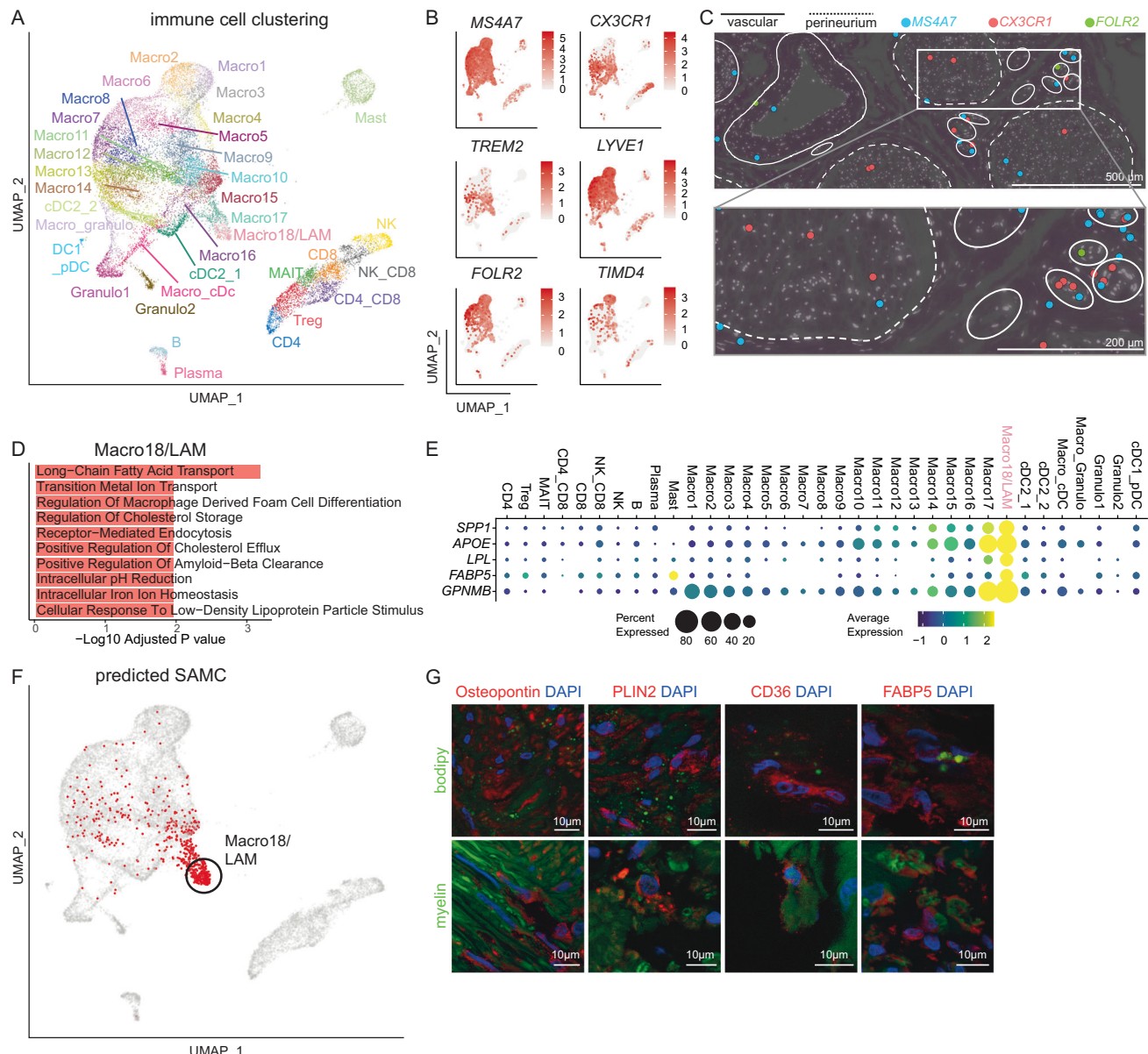

**Fig. 2 | Heterogeneity of human nerve-associated immune cells. A** UMAP of 18,436 nuclei representing 35 immune cell (IC) subclusters of 37 human sural nerves. **B** Feature plots of known endoneurial and epineurial macrophage marker genes. Color encodes gene expression. **C** Spatial-seq was performed on sural nerve samples from a total of eight patients. Representative spatial-seq images of the macrophage marker *MS4A7* and the endoneurial macrophage markers *CX3CR1* and *TREM2* in the sural nerve of CTRL patient S24. Each dot represents the expression of one transcript, a dotted line marks the perineurium, and a solid line surrounds individual vessels. **D** Gene ontology term enrichment analysis of marker genes (log₂ fold change > 2, adjusted *p*-value < 0.001) expressed by the Macro18/LAM cluster in

a one vs. all immune cell cluster comparison. Statistical significance was assessed using enrichR (one-sided Fisher's exact test with p-values adjusted for multiple comparisons using the Benjamini-Hochberg method). **E** Gene expression of lipid-associated macrophages in the immune cell subclusters. **F** Stroke-associated myeloid cells (SAMC) from Beuker et al. were projected onto the immune cell clustering. Each red dot denotes a nucleus predicted to be a SAMC.
**G** Representative histological section characterizing Macro18/LAM cluster by co-staining marker genes with myelin and BODIPY (labels neutral lipids). *SPP1* encodes osteopontin. Nuclei were stained with DAPI. Stainings were performed on a minimum of two sural nerve sections per patient from a total of three patients.

human tissue (Supplementary Fig. 3E). *Prima1* expression was not detectable in the murine nerve or positive control tissue. In contrast to rodent snRNA-seq, *Cxcl14* was detected in the murine perineurium using RNA-ISH. As expected, *Cxcl14* colocalized with the perineurial marker *Krt19* (Supplementary Fig. 3E). What appeared to be human specificity in snRNA-seq was therefore at least partly attributable to technical factors. Nonetheless, we provide multi-modal confirmation of several novel cell type markers: *MLIP* (myelinating Schwann cells), *GRIK3* and *PRIMA1* (non-myelinating Schwann cells), and *CXCL14* (perineurial cells).

## Diseased human peripheral nerves contain myelin-phagocytosing macrophages

We next deeply subclustered all immune cell (IC) nuclei (*n* = 18,436) at high resolution (Fig. 2A). In addition to the initial clustering, we identified multiple smaller myeloid clusters that could be assigned to classical dendritic cells (DC) type 1 (cDC1; *CLEC9A*, *XCR1*, *BATF3*), plasmacytoid DC (pDC; *CLEC4C*, *IRF8*), and classical DC type 2 (cDC2; *FCER1A*, *CD1C*, *CLEC10A*) (Fig. 2A and Supplementary Fig. 4A). cDC2 had not been previously described in nerves[13]. Nerve-associated T/NK cell clusters, naive B cells (*CD19*, *MS4A1*/CD20, *CD79B*), and plasma

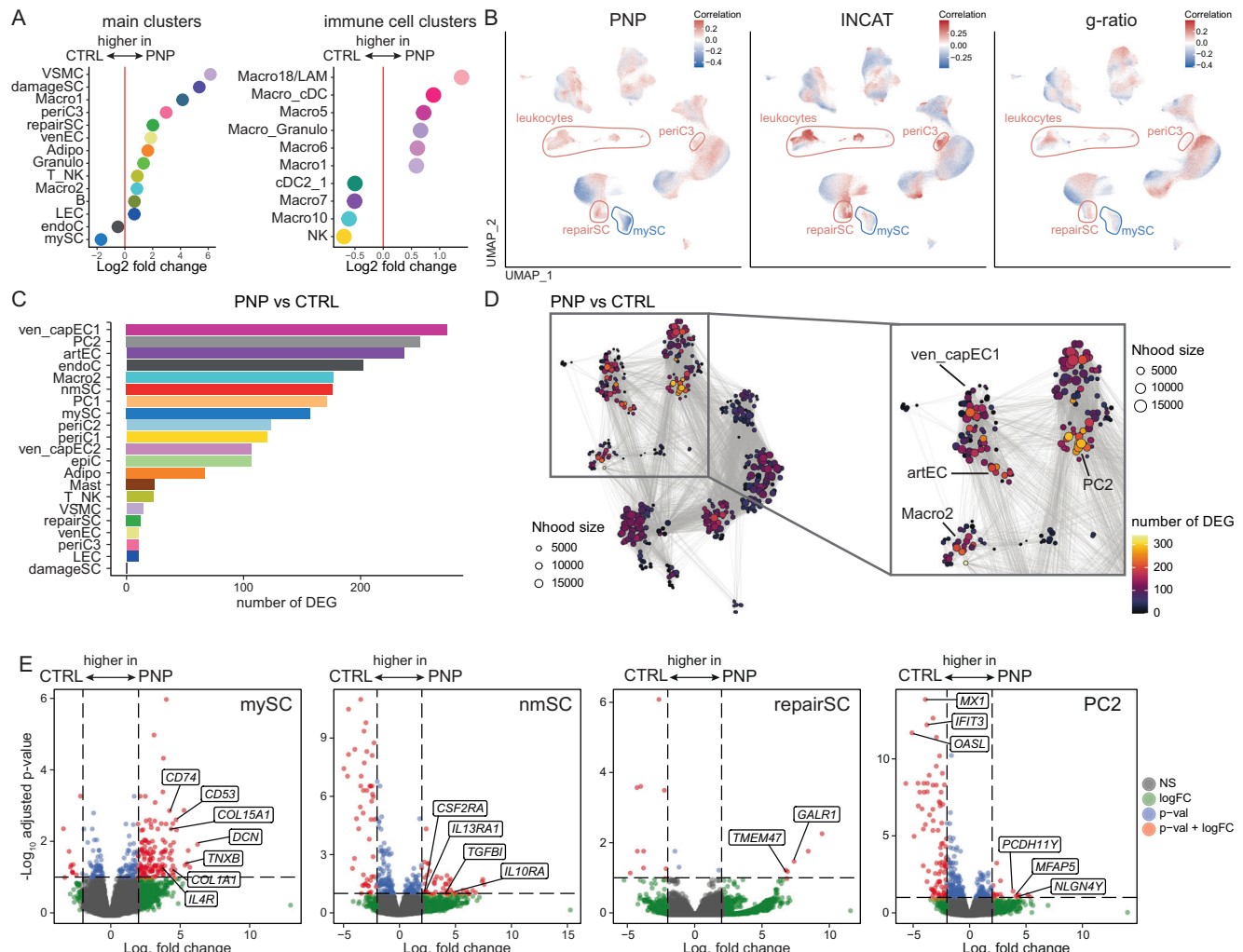

**Fig. 3 | Identifying a pan-neuropathy cellular transcriptional response pattern.** **A** Differences in cluster abundance in patients with polyneuropathy (PNP) vs. controls (CTRL), left: main clusters (Fig. 1B), right: immune cell subclusters (Fig. 2A). Only clusters with a log₂ fold change > 0.5 are shown. **B** Correlations between individual single nuclei and PNP status (right), Inflammatory Neuropathy Cause and Treatment (INCAT) disability score (middle), and g-ratio of myelinated axons (right) are shown. Red indicates a positive, blue a negative correlation. **C** The number of differentially expressed genes (DEG) per cell cluster was determined in a pseudobulk approach and is visualized. **D** Neighborhood graph illustrating the number of DEG between PNP and CTRL in a cluster-free neighborhood-based approach. Each dot denotes a neighborhood. Edges reflect the number of cells shared between neighborhoods. The panel on the right displays a higher magnification of the neighborhoods with the largest number of DEGs. **E** Volcano plots of DEG in 0.1, and the vertical dashed lines display a log₂ fold change of 2. Selected transcripts above those thresholds are labeled. Statistical significance was assessed using a two-sided likelihood ratio test, with p-values adjusted for multiple comparisons using the Benjamini-Hochberg method.

cells (*JCHAIN*, *SDC1*/CD138) were also detected (Fig. 2A and Supplementary Fig. 4A). B lineage cells expressed class-switched (*IGHG1*, *IGHG3*, *IGHG4*, *IGHA1*) heavy chain genes (Supplementary Fig. 4B). We thus delineated nerve-associated leukocytes in humans with high resolution.

The majority of IC clusters (79% of all IC nuclei) were of myeloid lineage, mainly macrophages (Macro1-18: *LYZ*, *CD14*, *MRC1*, *CD163*, *MS4A7*) (Fig. 2A, Supplementary Fig. 4A and Supplementary Data 8). The phenotype and ontogeny of nerve-associated macrophages in rodents differ depending on their endo- vs. epineurial location[8,21,22]. In our human dataset, macrophage clusters (*MS4A7*) also exhibited a gradient of expression from known endoneurial (*CX3CR1*, *TREM2*) to epineurial (*LYVE1*, *FOLR2*, *TIMD4*) markers (Fig. 2B). Location-specific markers of epi- (*FOLR2*) vs. endoneurial (*CX3CR1*) macrophages (*MS4A7*)[21] were accordingly confirmed in spatial transcriptomics (Fig. 2C). The phenotype of nerve-associated macrophages is thus conserved across species.

One representative tentatively endoneurial macrophage cluster (Macro18) was notably enriched in lipid metabolism-related pathways (Fig. 2D). Specifically, Macro18 showed high expression of *SPP1*, *APOE*, *LPL*, *FABP5*, and *GPNMB* (Fig. 2E), and thus a transcriptional phenotype similar to lipid-associated macrophages (LAMs)[23] and to a previously described CNS myeloid population in stroke with a phagocytic and lipid-sensing phenotype[24]. Accordingly, when projected onto our dataset, the stroke-associated myeloid cells (SAMC) predominantly aligned with the Macro18 cluster (Fig. 2F). Histologically, Macro18 markers (*SPP1*/Osteopontin, PLIN2, CD36, FABP5; Fig. 2E and Supplementary Data 5) colocalized with myelin and BODIPY, which labels neutral lipids in samples from PNP patients (Fig. 2G). Macro18 thus likely represents a population with myelin-phagocytizing capacity similar to myelin debris-clearing stroke-associated myeloid cells[24]. Damage response mechanisms in the presence of lipid excess may thus be shared between the central and peripheral nervous systems.

## Non-endoneurial cell types profoundly respond to polyneuropathy

We next systematically characterized how PNP affected peripheral nerves using the snRNA-seq data (Supplementary Fig. 1E). The most apparent change in PNP was a loss of mySC and the occurrence of damageSC and repairSC (Fig. 3A), which are known to be induced by nerve damage in rodents[15,25–27]. Among IC subclusters, the most prominent change in PNP patients was an increase in the Macro18 subcluster, indicative of endoneurial LAM (Fig. 3A) with a potential role in myelin debris clearance. In addition, the perineurial cell cluster periC3 also increased in PNP (Fig. 3A). When testing which individual single nuclei were associated with disease, the PNP status positively correlated with nuclei in the leukocyte and repairSC clusters and negatively correlated with the mySC cluster after correcting for confounders (Methods) (Fig. 3B, left panel). This means that mySC are less likely to be present in PNP patients on a per-nucleus level. In addition, the periC3 cluster positively correlated with PNP disease status (Fig. 3B, left panel). These clusters similarly correlated with clinical disease severity as quantified by the clinical INCAT score (Fig. 3B, middle panel) and myelin thickness (inversely measured by g-ratio) (Fig. 3B, right panel). PNPs thus induced profound compositional changes that correlated with disease severity and established measures of PNP, and they strongly affected perineurial cells.

We next characterized which and how nerve cells *transcriptionally* responded to the disease. The number of differentially expressed genes (DEG) was highest in vascular clusters (ven_capEC1, PC2, artEC), followed by endoneurial fibroblasts (endoC) (Fig. 3C). Using a cluster-free neighborhood-based approach, we accordingly found that neighborhoods with the most DEG were located in vascular clusters (PC2, ven_capEC1, artEC) and in the Macro2 cluster (Fig. 3D). Next, we more specifically tested how PNP influenced gene expression of selected nerve-associated cells (mySC, nmSC, repairSC, PC2) (Fig. 3E). We found that many of the DEG in the mySC cluster were associated with fibrotic tissue remodeling (e.g., *DCN, TNXB*), extracellular matrix formation (e.g., *COL1A1, COL15A1*), and immune regulation (e.g., *CD53, IL4R, CD74*). The repairSC cluster expressed genes for nerve repair and neuronal differentiation (*GALR1, TMEM47*) (Fig. 3E and Supplementary Data 9). GO term enrichment analysis showed that genes upregulated in PNP were associated with cell differentiation in the mySC cluster and with cell migration in the nmSC cluster (Supplementary Fig. 4C). In summary, PNPs induced transcriptional changes in Schwann cells but also widely outside of the endoneurium, especially in non-endoneurial vascular cells. PNPs thus surprisingly affected the entire cellular micro-milieu of peripheral nerves even beyond the endoneurium and may thus constitute 'pan-nerve diseases'.

## PNP subtypes preferentially affect different cellular compartments of peripheral nerves

We next sought to better understand the poorly defined heterogeneity of PNP mechanisms. We classified the available PNP patients by integrating all diagnostic information (Supplementary Data 1) into seven subtypes of PNPs: vasculitic (VN, $n = 5$), chronic inflammatory demyelinating (CIDP, $n = 9$), chronic idiopathic axonal (CIAP, $n = 11$), cancer-associated paraproteinemic (PPN, $n = 2$), diabetic (DPN, $n = 2$), other inflammatory (OIN, $n = 2$), and other non-inflammatory (ONIN, $n = 2$) (Fig. 4A). Histological characterization (Supplementary Fig. 5A–G) identified that myelin thickness (Supplementary Fig. 5B, C) and the average number of intactly myelinated axons (Supplementary Fig. 5D) decreased in PNPs, while axon diameter was less affected (Supplementary Fig. 5E). Histological findings in PNP subtypes were in accordance with expectations and correlated with electrophysiology (Supplementary Fig. 5F) and with cluster proportions determined by snRNA-seq (Supplementary Fig. 5G).

We next compared cluster proportions between groups with at least four samples per group (VN, CIDP, CIAP, CTRL), while disregarding others. In VN, CIDP, and CIAP, we observed a relative increase in damageSC and repairSC, alongside a reduction in mySC, compared to controls (Fig. 4B). Schwann cell pathology thus occurred widely across PNPs. A gain of the periC3 and the venEC cluster was shared between VN and CIDP. An increase in the VSMC cluster was found in both VN and CIAP. Loss of the LEC cluster was specific to VN (Fig. 4B). The lipid-associated Macro18/LAM immune cell cluster was increased in all three PNP groups compared to CTRL (Fig. 4C). Only VN showed an expansion of the plasma cell cluster (Fig. 4C). Compositional cellular analysis of PNP subtypes thus revealed unique and shared cellular alterations, including a shared increase of Macro18/LAM.

We next analyzed spatial transcriptomics of sural nerves from VN, CIDP, and CIAP patients vs. CTRL ($n = 2$ per group). Predicted repairSCs were increased in the endoneurium in VN (Fig. 4D). Predicted mySCs were considerably less abundant in VN than in CTRL (Fig. 4D), and this was also true but less prominent for CIDP and CIAP (Supplementary Fig. 5H, I). Quantification of leukocyte-associated transcripts (*PTPRC*/CD45, *MS4A1*/CD20, *CD3E*) showed that the density of endo- and epineurial B and T cells increased in VN, CIDP, and CIAP compared to CTRL (Fig. 4E and Supplementary Fig. 5J, K). Predicted T_NK cells were more abundant in VN and less so in CIDP and CIAP than in CTRL (Supplementary Fig. 5L, M). *TREM2*, as a marker expressed by Macro18/LAM (Fig. 2B), was predominantly detected in the endoneurium (Fig. 4F) and was increased in CIDP, CIAP, and most notably in VN compared to CTRL (Supplementary Fig. 5N). We thus spatially validated the gain of repairSC, leukocytes, and LAM along with the loss of mySC in PNPs, with the most prominent effects in VN.

Next, we tested for subtype-specific transcriptional alterations. In VN, most DEGs were located in neighborhoods in the PC2 (pericyte) cluster (Fig. 4G) in accordance with vascular-focused inflammation. In CIDP, DEGs were also identified in neighborhoods of vascular clusters (ven_capEC1, artEC, PC2) and in the Macro2 cluster. In CIAP, DEGs were distributed in a more widespread pattern (Fig. 4G). Each PNP subtype thus induced unique compositional and transcriptional responses.

## PNPs, particularly immune-mediated subtypes, exhibit focal perineurial hyperplasia

We next aimed to spatially validate the notable increase in the perineurial cell clusters in snRNA-seq. Plotting known (*CLDN1, SLC2A1, KRT19*) and novel (*CXCL14*) perineurial markers identified the perineurium as a single cell layer in CTRL but as 3–5 cell layers in PNP samples. This perineurial hyperplasia was most pronounced in CIDP and appeared to be regionalized (Fig. 5A). The perineurium was also more dispersed and heterogeneous in PNP samples (Fig. 5A). Moreover, the expression of the novel perineurial marker *CXCL14* was increased in PNPs compared to CTRL in snRNA-seq and immunofluorescence (Fig. 5B, C).

To validate the perineurial hyperplasia in CIDP, we histologically stained epithelial membrane antigen (EMA)[28] in 18 additional controls (residual material of surgical sural nerve autografts) and 12 CIDP patients (Supplementary Data 10) and measured the area and perimeter of the perineurium for each fascicle (370 fascicles in total) (Fig. 5D). We applied a linear mixed model to account for potential confounders (age and sex as fixed effects) and center- and sample-specific variability (random effects). We found that the perineurial area and perimeter were significantly increased in CIDP patients compared to CTRL (Fig. 5E). Moreover, their variance significantly increased in CIDP (Fig. 5F), suggesting that fascicular involvement was regionalized. In conclusion, CIDP exhibited focal perineurial hyperplasia driven by an expansion of PNP-specific perineurial cells, potentially induced by immune-perineurial *CXCL14* signaling.

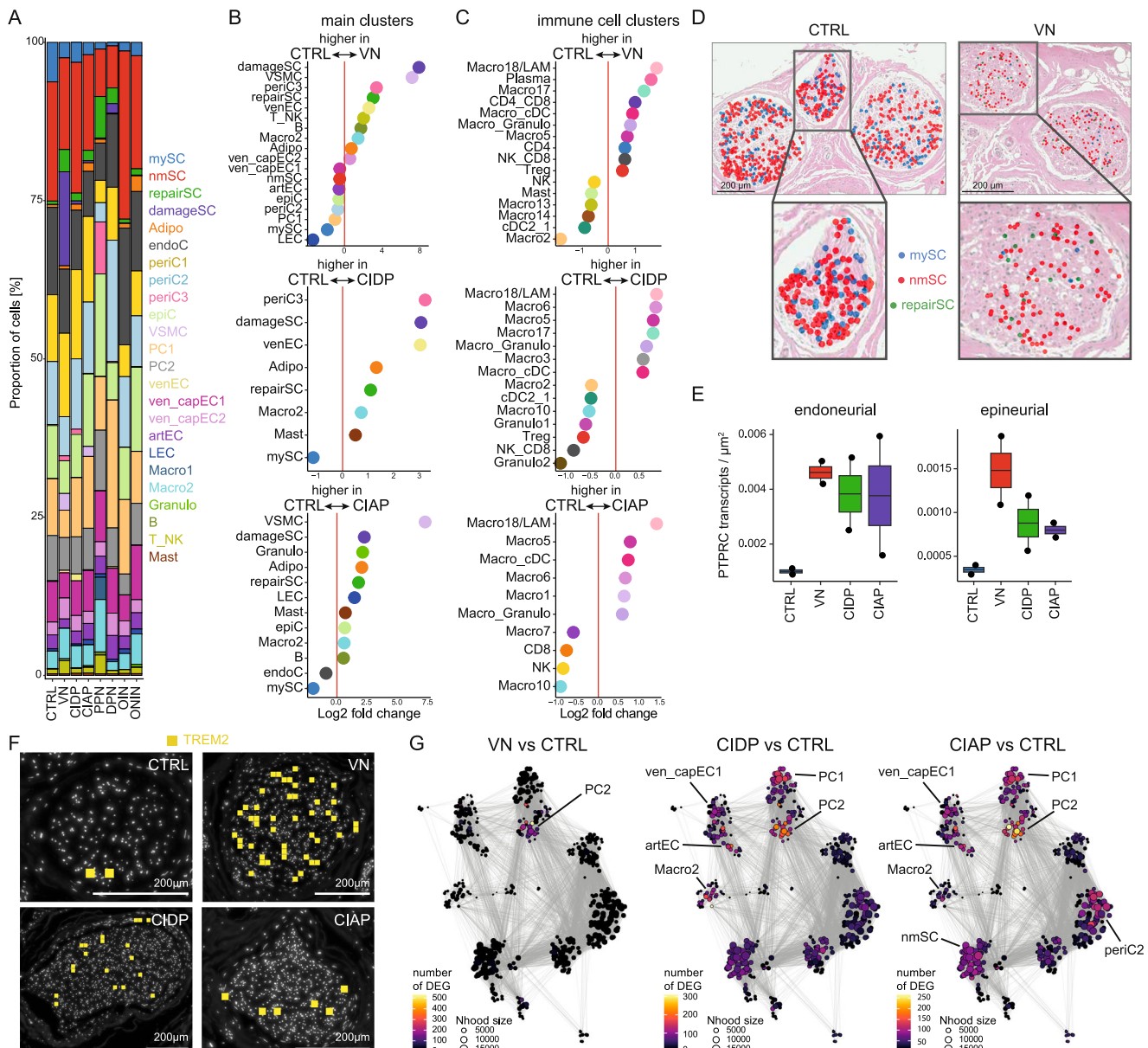

**Fig. 4 | PNP subtypes affect different cellular compartments of peripheral nerves. A** Proportion of cells split by disease group and colored by the main clusters. **B**, **C** Changes in cluster abundance in patients with vasculitic (VN), chronic inflammatory demyelinating (CIDP), and chronic idiopathic axonal (CIAP) polyneuropathy vs. controls (CTRL). The right panels display changes of the main clusters (as in Fig. 1B), the right panels show differences in the immune cell subclusters (Fig. 2A). Only clusters with a log₂ fold change > 0.5 are shown. **D**–**F** Spatial-seq was performed on sural nerve samples from a total of eight patients (two per disease). **D** Representative sections of H&E staining of sural nerves overlaid with spatial-seq showing predicted Schwann cell clusters in CTRL (S24, left panel) and

VN (S30, right panel) patients. Each dot represents one cell. **E** The density of *PTPRC* transcripts in the endoneurium (left pot) and epineurium (right plot) per disease group (*n* = 2 per disease group). Box plots show the median, interquartile range (IQR), and whiskers extending to 1.5 × IQR. Dots represent individual measurements. **F** Representative spatial-seq images showing *TREM2* transcripts (yellow) in the perineurium of CTRL (S24), VN (S30), CIDP (S01), and CIAP (S04) patients. Each square indicates the expression of one transcript. **G** Neighborhood graphs displaying the number of DEG between PNP patients with VN (left), CIDP (middle) or CIAP (right), and CTRL in a cluster-free neighborhood-based approach. Edges reflect the number of cells shared between neighborhoods.

## Discussion

We aimed to exploit the full potential of human nerve biopsies using state-of-the-art techniques and present a single-nucleus transcriptomics atlas comprising 365,708 nuclei of 33 PNP patients and four controls integrated with subcellular spatial transcriptomics. Capitalizing on this novel application to human peripheral nerves and a dataset 10-fold larger than previous rodent studies[8–13], we discovered and validated previously undescribed transcriptional markers of perineurial fibroblasts (*CXCL14*) and myelinating (*MLIP*) and non-myelinating SC (*GRIK3, PRIMA1*). In human PNPs, especially in immune-mediated forms, we identified perineurial hyperplasia

potentially driven by *CXCL14* signaling. Myelin-phagocytosing nerve and lipid-associated macrophages accumulated across PNPs, and transcriptional changes widely affected multiple non-endoneurial cell populations, suggesting PNPs as 'pan-nerve' diseases. In conclusion, we identified a PNP subtype-specific composition and transcriptome with diagnostic potential.

Our study has limitations: Our patient cohort as a whole is large in comparison to other single-cell transcriptomics studies[29], but limited patient numbers within each PNP subtype might be insufficient to fully explore differences between subtypes. Our results are potentially confounded by imbalances in disease duration, sex, center of origin,

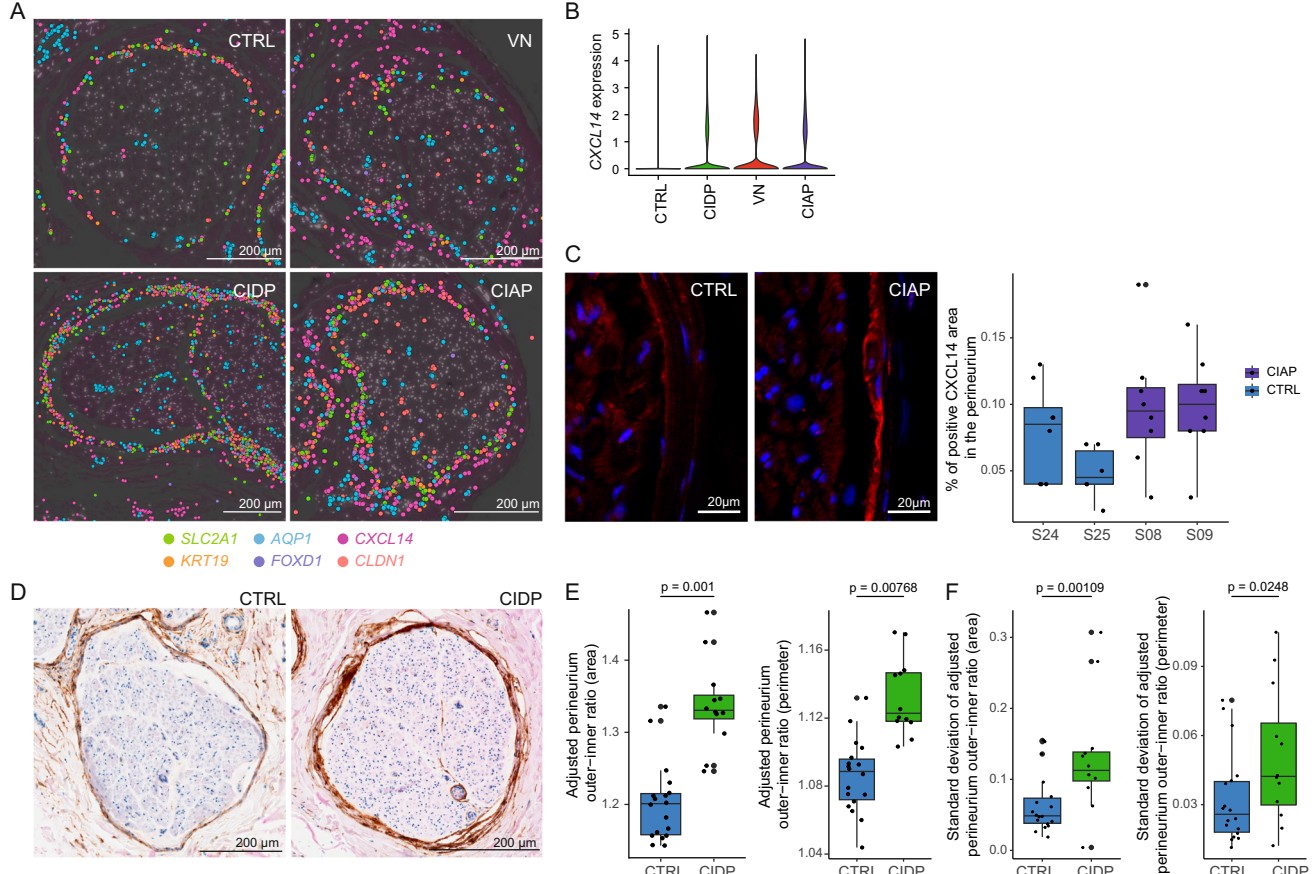

**Fig. 5 | PNPs, particularly immune-mediated subtypes, exhibit focal perineurial hyperplasia. A** Spatial-seq was performed on sural nerve samples from a total of eight patients (two per disease). Representative spatial transcriptomics images of perineurial marker genes of CTRL (S24), VN (S30), CIDP (S01), and CIAP (S14). Each dot indicates the expression of one transcript. **B** Average gene expression of *CXCL14* expression per disease group in snRNA-seq. **C** Representative immuno-fluorescence images showing CXCL14 staining in CTRL (S24) vs. CIDP (S08) (left panel). Immunofluorescence staining was performed on a minimum of six sural nerve sections per patient from a total of four patients. The quantification of the CXCL14-positive area in the perineurium is visualized in a boxplot (right panel). Box plots show the median, interquartile range (IQR), and whiskers extending to 1.5 × IQR. Dots represent individual measurements (technical replicates; S24/S08/S09: $n = 8$, S25: $n = 6$). **D** EMA staining was performed on sural nerve samples from a total of 30 patients. Representative images of epithelial membrane antigen (EMA) staining in CTRL (S71) and CIDP (S76). **E, F** The inner and outer perimeters, along with the corresponding area, were measured in CTRL ($n = 18$) and CIDP ($n = 12$) patients. A linear mixed model was used to adjust for confounding variables (Methods) and to determine statistical significance (two-sided $t$ test). Box plots show the median, interquartile range (IQR), and whiskers extending to 1.5 × IQR.

and immunosuppressive therapy in some patients (Supplementary Data 1). Larger multi-centric, multi-national funding, collection, and analysis efforts will be required in the future. Sural nerve biopsies are invasive with some inherent risk and are usually performed in difficult-to-diagnose patients[7]. Our patient cohort is thus biased towards rare PNP etiologies and, for example, includes only two diabetic PNP, the most common cause of PNPs[1,4,6]. We mitigate this by focusing on groups with at least four patients and analytically correcting for biases when possible. We believe that novel techniques to enrich rare cell types[30] could identify PNP-associated cellular patterns in more readily available tissues, such as skin biopsies, in the future. Whether less expensive and less labor-intensive methods, such as bulk transcriptomic or methylome analysis of peripheral nerves, could be similarly powerful remains to be determined.

Among novel cell type transcripts, the *MLIP* gene encodes a muscular laminin-interacting protein required to maintain muscular integrity[31], which could serve similar functions in SC. The *PRIMA1*-encoded protein organizes esterases at the neuromuscular junction in terminal SC[32] but has not been described in nmSC. *GRIK3* encodes a glutamate ionotropic receptor[33] and variants in *GRIK3* have been associated with hereditary neuropathy[34]. These genes may thus be involved in the structural organization of SC in human peripheral

nerves. Human *CXCL14* (synonymously *BRAK*) is constitutively expressed by epithelial tissues and regulates microglial development in the brain[35] and neurovascular patterning in the eye[36]. It is thus conceivable that the perineurium, regarded as an epithelial barrier[37], expresses *CXCL14* and thereby contributes to nerve organization. Upregulation of *CXCL14* was previously described in a mouse model of hereditary neuropathy[38]. Here, we found an increase in *CXCL14* expression in perineurial cells in human PNPs, suggesting *CXCL14* as a potential disease marker. The predicted interaction of *CXCL14* with B cell-derived *CXCR4* signals hints towards immunity-driven perineurial hyperplasia.

The perineurium is a lamellated structure made up of concentric cell layers bordered on each side by a basement membrane[39]. The perineurium forms an active diffusion barrier to maintain the homeostasis of the endoneurium[40]. A linear relationship has been described between fascicle diameter and perineurium thickness[41], i.e., larger fascicles have a thicker perineurium. In addition, distal sections have a greater perineurium thickness than proximal segments[41]. Perineurial thickening is a characteristic of diabetic PNP[42] and Borrelia neuropathy[43]. Focal or generalized perineurial thickening also profoundly occurs in 'perineuritis', a rare inflammatory disease[44]. Proliferation of PNP-specific perineurial cells could reflect a maladaptive

response of the nerve microstructure to chronic damage or inflammation. Overall, perineurial thickening per se has not been studied systematically in PNPs and may be diagnostically more relevant than previously appreciated.

Conditions such as obesity, stroke, and atherosclerosis induce tissue damage accompanied by local lipid excess and trigger the accumulation of lipid-associated macrophages (LAM) in the affected tissues[23,24,45]. LAMs share a common set of lipid-metabolizing transcripts (e.g., *Fabp5, Lpl, Spp1, Trem2*) and likely represent an adaptive response aimed at clearing excess lipids from the site of injury. Nerve-associated macrophages feature complex location-specific ontogeny[21] and are implicated in multiple nerve diseases and in nerve regeneration[46]. By defining LAMs in the peripheral nerves of human PNP patients, we here identified that the innate immune response to damage is conserved across lipid-rich tissues and across diverse injury conditions. This may hold even wider implications for PNPs, given that the lipid-sensing TREM2/APOE pathway on disease-associated macrophages (DAM) constitutes a potential biomarker and preclinical therapeutic target of neurodegenerative CNS diseases[47,48].

## Methods

### Collection of patient samples
We collected sural nerves from 33 PNP patients from three centers: University Hospital of Münster (10 patients), Essen (11 patients), and Würzburg (12 patients). In addition, we obtained sural nerves from four control patients in Essen with traumatic nerve injuries who received a sural nerve autograft as interposition but were unaffected by polyneuropathy (Supplementary Data 1). Residual material from the sural nerve grafts was used for control samples in this study. In PNP patients, sural nerve biopsy was performed as part of the clinical diagnostic workup, and a small portion of the nerve (approximately 0.2 cm) was used for this study. In Münster and Essen, the samples were immediately fresh-frozen after surgery in dry ice-cooled methylbutane and stored in liquid nitrogen (0–14 months, Supplementary Data 1) until nuclei extraction. In Würzburg, the samples were embedded into an OCT-embedding matrix (Roth) until nuclei extraction (33–77 months, Supplementary Data 1). In Essen and Münster, samples were collected from newly recruited patients. In Würzburg, samples were collected from an existing biobank of OCT-embedded cryo-preserved sural nerve biopsies collected between 2016 and 2020. All experiments were carried out in accordance with the Declaration of Helsinki and were approved by the local ethical committees in Münster (2018-719-f-S), Essen (21-10376-BO), and Würzburg (238/17; 15/19). All patients gave written informed consent to sample collection.

### Patient characteristics used for sequencing
Patients were characterized clinically regarding diagnosis, disease activity, therapy response, disease duration, immunosuppressive therapy, relevant secondary diagnosis, electrophysiological studies, and CSF analysis, outlined in Supplementary Data 1. Patients were diagnosed according to the EAN/PNS criteria 2021 for CIDP[49] and according to the PNS criteria for VN.

### Nuclei extraction and purification
For each patient, approximately 10 mg pieces (median of 13.9 mg) were cut on dry ice as starting material from the original samples. Nerve pieces were cut into smaller pieces and processed using the Miltenyi Biotec nuclei extraction protocol. In summary, samples were transferred to a gentleMACS C-tube (Cat. no. 130-093-237) containing 2 ml lysis buffer (nuclei extraction buffer Cat. no. 130-128-024 + 0.2 μ/μl RNase inhibitor EO0381) and processed on a gentleMACS dissociator with program 4C_nuclei_1. After dissociation, the nuclei suspension was filtered through a pluriStrainer Mini 70 μm cell strainer

(pluriSelect®) and washed in 200–400 μl resuspension buffer (PBS with 0.1% BSA and 0.2 μ/μL RNase inhibitor EO0381), depending on sample size and expected nuclei count. Nuclei suspension was then filtered through a 40 μm cell strainer (pluriSelect®), placed on a 1.5 ml DNA-LoBind tube (Merck). The nuclei suspension was stained with Trypan Blue to manually assess nuclei viability and count using a Fuchs-Rosenthal chamber. Equal volumes between samples were used for the downstream application of single-nucleus RNA seq.

### Single-nucleus RNA sequencing and generation of count matrices
Single-nucleus suspensions were loaded into a Chromium Next GEM Chip G and placed into the Chromium X Single Cell Controller and processed with Chromium Next GEM Single Cell 3´ Kit v3.1 reagents (all 10X Genomics). Sequencing was performed on Illumina Nextseq 2000 and Novaseq 6000 with a 28-8-0-91 read setup. We used cellranger v7.0.1 (10X Genomics) to generate count matrices with default parameters but an optimized transcriptome reference v1.1[50] for GRCh38.

### Single-nucleus analysis
CellBender v0.3.0[51] was used to remove background noise. In CellBender, the number of expected cells and total droplets was based on the UMI curve, and the learning rate was reduced based on the automated output report. Further downstream analysis was performed with the R package Seurat v5.0.1[52]. To increase computational efficiency and reduce the memory usage of this large dataset, we used BPCells v0.1.0. Low-quality cells were filtered for each sample individually by inspecting quality control plots and removing cells with higher mitochondrial percentages (range: 1–5%), low (< 200) or very high molecule counts per cell (range: 6000–9000). Doublets were removed using scDblFinder v1.16.0[53] with default parameters. We used Seurat for normalization (LogNormalize, default parameters), identification of highly variable genes (vst method, 2000 features), scaling, and performing PCA (default parameters). Next, we used atomic sketch integration (method LeverageScore, 5000 cells) in Seurat to reduce memory usage. Batch effects were accounted for by integrating the samples with scVI v1.0.4[54]. After inspecting the effect of batch removal, the full dataset was integrated with Seurat. The scVI integrated full dataset was used to calculate the UMAP embeddings with Seurat (30 dimensions). We identified clusters with the FindNeighbors and FindClusters (resolution 0.7) functions. We calculated the top-expressing genes of each cluster with FindMarkers (min.pct = 0.1, logfc_threshold = 0.25, p_val_adj < 0.05, two-sided Wilcoxon rank-sum test). To subcluster the immune cells, we selected the immune cell clusters of the main clusters. We then performed identification of highly variable genes, scaling, and PCA as explained above. The data were integrated with reciprocal PCA using Seurat, and after plausibility checks of the batch removal, UMAP was calculated (30 dimensions). Clusters were identified with FindNeighbors and FindClusters (resolution 2.3). The top-expressing genes were identified as explained for the main clusters. Enrichment analysis of top markers (avg_log2FC > 1, p_val_adj < 0.001) was carried out with enrichR 3.2[55] with the GO Biological Process 2023 database. Differentially expressed genes between conditions were identified using a pseudobulk method with Libra v1.0.0[56] (edgeR 4.0.7[57], two-sided likelihood ratio test). Volcano plots of DE genes were created with Enhanced Volcano v1.20.0. In addition, we determined DE genes in a cluster-independent manner with miloDE v0.0.0.9[58] following the tutorial. Briefly, neighborhoods were assigned based on the scvi integrated data (k = 30, prop = 0.1, d = 30), and DE testing was conducted between conditions (min_count = 10). To determine the cluster abundance, we used propeller[59] (part of speckle v1.2.0). We used the co-varying neighborhood analysis (CNA), implemented in the rcna package v0.0.99[60], to compute correlations between clinical phenotypes or histological measures

and our single-cell data independent of clustering, controlling for age and sex. To analyze cell-cell interactions, we used LIANA v0.1.14[61] based on the methods NATMI, Connectome, SingleCellSignalR, and logFC.

## Comparison with published datasets

We downloaded the publicly available annotated data from Yim et al.[9] (GEO GSE182098, sciatic nerve), Gerber et al.[10] (https://snat.ethz.ch/seurat-objects.html, 10X Genomics P60), Mathys et al. (https://compbio.mit.edu/scBBB/ ROSMAP vascular cells)[14], Wolbert et al.[8] (GEO GSE142541, mouse), and Beuker et al. (GSE189432). If necessary, the data were preprocessed with Seurat, i.e., normalized with Log-Normalize, highly variable genes were identified and scaled, and PCA was performed with default parameters. To identify novel marker genes, cell markers in the published datasets were determined with the FindMarkers function in Seurat (min.pct = 0.1, logfc.threshold = 0.25, p_val_ad < 0.05). To annotate our dataset based on the published dataset, rodent gene names were converted to orthologues using homologene v.1.4.68 (homologeneData2 database). We then classified our cells based on the annotated reference dataset using Find-TransferAnchors and TransferData functions in Seurat with default parameters. Cells were labeled as unknown if the Seurat prediction score was below 0.3.

## Histology: Semi-thin sections and analysis

After biopsy, 1/3 of the sural nerve was embedded in epoxy resin (Fig. 1A, experimental setup). Semithin sections (1 μm) of sural nerves were cut using an EM UC7 ultramicrotome (Leica Microsystems) and stained with toluidine blue. The sections were scanned with a slide scanner (Münster/Würzburg: Grundium Ocus40; Essen: Zeiss Axio Scan Z.1, Hitachi HV F203SCL camera). After blinding, the entire number of total myelinated axons per sural nerve was counted manually by one investigator using the CellCounter plug-in of ImageJ v1.36. Physiologically unmyelinated axons (diameter < 1 μm) and Remak bundle fibers were not included.

The ImageJ g-ratio plugin was used to determine the myelin thickness and axonal diameter of 15% of the total myelinated axons per sural nerve. In randomly chosen myelinated axons, the inner and outer rim of a myelin sheath were manually traced by one investigator. The g-ratio was calculated by dividing the axonal circumference (i.e., inner rim of the myelin sheath) by the outer circumference of the respective myelin sheath, presuming circular axons. The g-ratio values of each patient were plotted against the calculated axon diameters.

## Preparation of sural nerve cross sections and spatial transcriptomics

We created a custom, standalone gene panel for spatial transcriptomics using the commercial Xenium Analyzer platform. The panel was designed based on our single-nucleus analysis, mostly focusing on cells that demonstrated a potential role in polyneuropathies, including Schwann cells (mySC, nmSC, repairSC), perineurial cells, and vascular cells such as pericytes and endothelial cells. We included a minimum of four top marker genes of each of the clusters and also defined twenty DE genes that were differentially expressed between the different PNP subtypes. We additionally included known markers of both epi- and endoneurial cells and different subtypes of leukocytes, including B cells, T cells, and macrophages, adding up to a total of 99 genes (Supplementary Data 7).

In total, 8 sural nerve formalin-fixed paraffin-embedded (FFPE) samples were included for Xenium spatial transcriptomics: 2 CTRL (S22, S24), 2 CIDP (S01, S11), 2 CIAP (S04, S14), and 2 VN (S29, S30) samples (Supplementary Data 1). Samples were primarily selected based on their optimal/intact morphology in the semithin sections and their center, with the aim to limit center bias.

Sample preparation and processing were carried out by the CMCB Technology Platform Core Facility EM and Histology, and the DRESDEN-concept Genome Center at the Technical University Dresden, in Germany. Samples were processed according to manufacturer protocols (10x protocols, CG000580, CG00582 and CG00584). Briefly, 4 μm cross sections were collected, floated in a 37 °C water bath, and adhered to Xenium slides (10x Genomics, PN 1000460). Samples were deparaffinized in 2x xylene and rehydrated in a descending series from 100% ethanol to MilliQ water. After inserting slides into Xenium cassettes, samples were decross-linked and incubated overnight (22 h) with padlock probes, followed by a post-hybridization wash. Subsequently, padlock probes were ligated, followed by rolling circle amplification, autofluorescence quenching, and nuclear staining (DAPI). Slides were loaded on a 10X Xenium Analyzer for region selection, with each region corresponding to one entire nerve bundle.

Primary analysis, image processing and decoding, and secondary analysis, cell segmentation, and transcript mapping were performed on the instrument with the analysis software Xenium v1.6.0.8, resulting in a cell-feature matrix and an initial clustering. For cell segmentation, a custom neural network on DAPI images was initially used for nucleus segmentation. Then nucleus boundaries are expanded by 15 μm or until they encounter another cell boundary in X-Y to define cell boundaries. Transcripts were mapped to these 2D shapes according to their X and Y coordinates. Gene expression mapping was visualized using the Xenium Explorer software v.1.3. The post-Xenium H&E staining was processed according to the manufacturer protocol, including quencher removal in advance (see 10x Genomics protocol CG000613/A).

## Spatial transcriptomics data analysis

Xenium data was loaded into Seurat. We removed cells with less than 10 molecules per cell. We performed normalization with SCTransform (default parameters) and computed PCA for each sample. To match our snRNA-seq data to our Xenium data, each cell was classified with the FindTransferAnchors and TransferData function based on our previously annotated snRNA-seq data (downsampled to 1000 cells per cluster). Cells with a prediction score below 0.3 were labeled as unknown. The 24 main clusters were then aggregated into 8 larger groups (SC, endoC, periC, epiC, VSMC, PC, EC, IC). For manual quantification of endo- vs. epineurial transcripts, the Xenium images were loaded into Xenium Explorer v1.3.0 (10x Genomics) and endoneurial areas were manually outlined using the selection tool while using visualization of Schwann cell marker transcripts (*EGR2, NGFR, SOX10, S100B*) to reliably identify such endoneurial areas. The area of each image visibly unoccupied by nerve-associated tissue was selected as epineurium. The area (in μm²), the number of total segmented cells, and the number of immune cell-associated transcripts (leukocytes: *PTPRC*/CD45, B cells: *MS4A1*/CD20, T cells: *CD3E*) were quantified in each selection area. The density of the respective transcript per total endoneurial and epineurial area was calculated for each sample. For the *TREM2* quantification, the entire nerve area (in μm²), the number of total segmented cells, and the number of *TREM2* transcripts were quantified using the selection tool in the Xenium Explorer. Selected transcripts were visualized in Xenium Explorer. Images were integrated with corresponding H&E staining to visualize the spatial orientation.

## Combined immunofluorescence and RNA in situ hybridization (RNA-ISH) in murine tissue

10–15 week old female mice were perfused with phosphate-buffered saline (PBS) and sacrificed. The sciatic nerve and the brain were isolated and directly frozen in Tissue-Tek O.C.T.™ compound (Sakura Finetek Europe). 10 μm sections were cut at −20 °C using a cryostat (Epredia CryoStar NX50). Samples were processed according to the

HCR™ RNA-FISH protocol for frozen or fixed frozen tissue sections and the multiplexed HCR™ RNA-FISH protocol from Molecular Instruments, followed by immunofluorescence staining. Probes *Krt19* (B1; AB472075.1), *Cxcl14* (B2; NM_019568.2), and *Grik3* (B2; NM_001081097.3) were used to detect target RNA within intact cells, and the primary antibody mouse anti-L1cam (1:100, Abcam, ab24345) and AF488-labeled secondary antibody (1:100, Life, A-11001) were used for the immunofluorescence staining. Briefly, the fresh frozen samples were fixed in 4% paraformaldehyde for 15 minutes at 4 °C and then dehydrated in a series of 50%-70%-100% ethanol each for 5 min at room temperature. Samples were washed with PBS and incubated with a hybridization buffer for 30 min at 37 °C. A concentration of 0.5 pmol of each probe was used and incubated at 37 °C overnight. Excess probes were removed by applying a series of 25%-50%-75%-100% 5xSSCT (5x sodium chloride, sodium citrate (SSC) and 0.1% Tween 20) for 15 min each at 37 °C, except for the final 100% 5 x SSCT, which was incubated for 15 min at room temperature. Samples were pre-amplified in a humidified chamber for 30 min at room temperature and incubated with a hairpin solution (4.5 pmol/hairpin) for the respective probes overnight at room temperature. The next day, slides were washed with 5 x SSCT, and the immunofluorescence staining was performed. Samples were blocked using a blocking reagent for 15 minutes at room temperature, followed by incubation with the primary antibody for 1 h at room temperature. Subsequently, samples were incubated with a fluorescent-labeled secondary antibody for an additional hour at room temperature. Sections were mounted in a fluorescence preservative mounting medium containing DAPI (4',6-diamidino-2-phenylindole, Fluoromount-G, Invitrogen 00-4959-52), and images were obtained with a Zeiss AxioVision Apotome (Carl Zeiss) and processed in ZEN 2.3 lite (blue edition).

## Immunohistochemistry of human tissue

Formalin-fixed, paraffin-embedded nerve sections (5 μm) were rehydrated and antigen retrieved using EnVision™ FLEX Target Retrieval Solution (pH 6.0). After washing (PBS, pH 7.4), sections were incubated in blocking reagent (Roche, Mannheim, Germany) for 20 min (room temperature) and stained with the following primary antibodies: rabbit anti-Osteopontin (1:100, Abcam, ab63856), goat anti-FABP5 (1:100, R&D Systems, AF1476), rabbit anti-Perilipin-2 (1:100, Novusbio, NBP2-48532), rabbit anti-CD36 (1:100, Abcam, ab124515), and rabbit anti-CXCL14 (1:100, Abcam, ab264467). Myelin was stained with FluoroMyelin Green (1:100, 20 min, Thermofisher, F34651). All nerve sections were then incubated with the appropriate AlexaFluor secondary antibody for target antigen visualization (1:100, 45 min, room temperature). For lipid droplet and CXCL14 staining, frozen sural nerve tissue was first fixed in paraformaldehyde (pH 7.4, 4 °C) overnight, soaked in sucrose (30%, 4 °C) for up to 3 days, embedded in TissueTek, and cut with a cryostat (Epredia CryoStar NX50). Transverse nerve sections (10 μm) were blocked with blocking reagent (20 min, room temperature) and stained with the above primary antibodies. Lipid droplets were visualized using BODIPY 493/503 (1:1000, Molecular Probes, D3922). Cell nuclei were counterstained with a fluorescence preservative mounting medium containing DAPI (4',6-diamidino-2-phenylindole, Fluoromount-G, Invitrogen 00-4959-52). Fluorescence images were taken using a Zeiss AxioVision Apotome (Carl Zeiss).

To quantify CXCL14, images were acquired at 20x magnification with a consistent exposure time. Up to eight images of up to eight perineurial regions were taken per patient (two CTRL and two CIAP patients). ImageJ v.1.54 f was used for analysis. The perineurium was manually outlined to determine its total area, and CXCL14-positive regions were identified using a color threshold with a brightness value of 100. The CXCL14-positive area was measured, and the percentage of CXCL14-positive area within the perineurium was calculated relative to the total perineurial area.

## Perineurial thickness measurements

To quantify the thickness of the perineurium, we identified sural nerve histologies from 12 CIDP and 18 CTRL patients from three centers: University Hospital of Münster, Essen, and Leipzig (Supplementary Data 10). CTRL samples were residual material from sural nerve grafts. CIDP patients fulfilled the EAN/PNS criteria 2021 for CIDP[49].

In Münster, immunohistochemical detection of epithelial membrane antigen (EMA) was performed using an EMA antibody (Dako, clone E29, undiluted) on an automated Dako Omnis staining system. In Leipzig, automated immunostaining with the Benchmark Ultra Staining System and a primary antibody against EMA monoclonal antibody (Dako, clone E29, 1:50) were used. The slides were digitized using a Grundium Ocus 40 slide scanner microscope.

The area and the perimeter of the perineurium were manually measured for each fascicle in ImageJ v1.54 d by manually outlining the inner and outer perimeter of the perineurium of each intact nerve fascicle in images of EMA-stained sural nerve biopsies by an investigator blinded towards patient identity. The area occupied by the perineurium was calculated as the difference between the outer area and the inner area. The ratio of the areas and perimeters was calculated and used for the analysis.

To control for potential confounders, we used a linear mixed model with lme4 v1.1-35[62], including age and sex as fixed effects and center and sample ID as random effects. The model was employed to determine the significance between the CIDP and CTRL groups and to adjust the perineurium measurements. To test for variance heterogeneity, we calculated the standard deviation of the adjusted perineurium measurements. We built an additional linear mixed model with lme4 v1.1-35, including age and sex as fixed effects and center as a random effect to assess the significance of the standard deviation between the CIDP and CTRL groups.

## Reporting summary

Further information on research design is available in the Nature Portfolio Reporting Summary linked to this article.

## Data availability

The raw and processed single-cell sequencing data with sample and cluster annotations and spatial data generated in this study have been deposited in the GEO database under accession code GSE285983 and GSE 285984. An interactive version of the snRNA-seq data, created with cerebroAppLite v1.5.2[63] and an interactive version of the Xenium data, created with TissUUmaps v3.1[64], are available at https://pns-atlas.mzhlab.com. Source data are available on Zenodo (https://doi.org/10.5281/zenodo.14226218)[65].

## Code availability

The code is publicly available at https://github.com/mihem/pns_atlas). and has been archived on Zenodo (https://zenodo.org/records/15750104)[66]. A Docker image is available to restore the coding environment. Figures are reproduced in a markdown document that automatically downloads all relevant source data from Zenodo[65] available at https://mihem.github.io/pns_atlas/.

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

## Acknowledgements

We are deeply indebted to the patients for their participation. We thank Rebecca Ley from the Institute of Neuropathology Münster, Birgit Schmeddes and Sophie Linnenbaum from the Department of Neurology Münster, Susanne Weiche from the Dresden-concept Genome Center, and Kristina Wagner from the Department of Neurology in Essen for technical assistance. This work was supported by the Core EM and Histology, a core facility of the CMCB Technology Platform at the Technische Universität Dresden. Part of the calculations were performed on the high-performance computing (HPC) cluster PALMA II of the University of Münster, subsidized by the DFG (INST 211/667-1). This project was supported by the National Competence Network for Immune-Mediated Neuropathies in Germany (Kompetenznetz Peripherer Nerv). This project was mainly funded by a grant from the Bundesministerium für Bildung und Forschung (BMBF) 'Lipid Immune Neuropathy Consortium' (to G.M.z.H., M.S., R.S., R.F.) and a grant from the Interdisziplinäres Zentrum für Klinische Forschung (IZKF) Münster (SEED/016/21 to M.H.). In addition, GMzH was supported by grants from the Deutsche Forschungsgemeinschaft (DFG) (ME4050/12-1, ME4050/13-1, ME4050/8-1). This project was also supported by the DFG Sonderforschungsbereich Transregio 128 (to H.W.). J.A. was supported by the DFG Research Infrastructure NGS_CC (INST 269/768, project 407482635, DRESDEN-concept Genome Center). Open Access funding enabled and organized by Projekt DEAL.

## Author contributions

M.H. and G.M.z.H. conceived and supervised the study. M.H., J.T., A.K.M., M.M., M.S., A.-K.U., C.D., K.K., F.S., B.E., K.D., N.Ü., and C.S. were involved in collecting human sural nerves. J.T. performed nuclei extraction and snRNA-seq experiments. I.-N.L. carried out sequencing of snRNA-seq. J.W. planned the Xenium analysis. J.A. and A.D. performed Xenium experiments. F.D. determined g-ratios. M.M. measured the perineurium thickness. M.G., Christian T. and Carolina T. supervised perineurial stainings. M.H., N.G., and C.S. gathered sample and patient metadata. A.-L.B. and J.-K.S. performed RNA-ISH/IF and IH experiments. M.H. performed computational and statistical analyses. A.-L.B. designed the figures and schemes. J.M., R.S., R.F., H.W., C.S., and M.S. co-supervised the study. M.H., A.-L.B., J.W., and G.M.z.H. wrote the manuscript. All authors read and approved the manuscript.

## Funding

## Competing interests

M.H., J.W. and G.M.z.H. have submitted a patent application for the diagnostic classification of PNP patients by single-nucleus transcriptomics of human nerve material. The remaining authors report no competing interests.
