## [Transparent Peer Review file · Nature Communications]

Multi-omic identification of perineurial hyperplasia and lipid-associated nerve macrophages in human polyneuropathies

Corresponding Author: Professor Gerd Meyer zu Hörste

Version 0:

Reviewer comments:

Reviewer #1

(Remarks to the Author)

This study performs fairly large scale spatial transcriptomics on nerve tissue obtained from humans with polyneuropathies and several control tissue samples. Further analysis of the transcriptomic data reveal expected, and in some cases unexpected, cellular diversity in the non-neuronal components in the nerve. The authors note there is a complex immune cell profile in the nerve and find a loss of mySC in PNP. In addition to transcriptional changes in Schwann cells, the authors provide several novel cell type transcripts and are able to take all their observations and use them to create a tool/methodology that would potentially have diagnostic potential.

At heart, this is a descriptive study and a potentially useful one for both basic scientists and those clinically oriented. As far as I can tell, the transcriptomics are well done and reported rigorously. I have a few comments and thoughts about how the manuscript could be made most accessible to the larger community.

1. There are several references throughout the study of this being the "first" study to do this type of work. While this may be a matter of preference, I do not see the value of stating this as I think the study would be just as valuable if it were the second or third. Similarly, there are a references to this also being 'large-scale'. I worry that this is also a somewhat subjective statement. It is likely that in the future 'larger' scale studies would be done and therefore I think this is not that useful a feature.

2. The authors also make several references to the 'human-specific' nature of some of their findings. This is quite interesting, however, I am a little confused by this claim. If in fact the mouse datasets are simply smaller/less developed at the moment, could it not be that we simply have not had the resolution to see these markers in the mouse/rodent? It could be useful to have some form of comparison, computational or experimental, comparing the rodent and human dataset. To be clear, if these analysis reveal that in fact the rodent does in fact have the 'human-specific' markers, I do not view this as a problem or diminishing my enthusiasm for the study. I simply ask just to provide some clarity for the readers.

Reviewer #2

(Remarks to the Author)

This manuscript represents an impressive and substantial undertaking in which multiple omics approaches were used to analyze human healthy and diseased peripheral nerve biopsy samples. The utility and importance of these datasets for the research community is extremely high and needed- this will be an important resource for many different studies in the coming years. In general, a weakness of the paper is its descriptive nature, which is largely inherent to the type of data generated. Some thoughts on how this might be addressed:

Major criticisms:

1. In Figure 1C and E, CXCL14 and other novel markers of perineurial, BNB and Schwann cells are highlighted. Can the authors use additional computational approaches to understand expression pattern of CXCL14's binding partner and how this interaction may inform cell-cell interactions?
2. What is the potential pathogenic role of Cluster 18, the cell type highlighted in Fig 2 and 3? Are lipid metabolism pathways upregulated due to myelin phagocytosis?
3. Can the authors provide evidence that the MVRI approach in Fig 5 is a valid way to group patients? This "hypothesis free"

testing is proposed as a potential diagnostic, so proving it is valid on another dataset is important.

Minor points:

1. Suppl is labeled Suppl Fig 5
2. This sentence related to Fig 5 requires editing: Integrating histological measures into the PCA analysis did not show evident enrichment of a certain histological feature in the PCA space.(Suppl. Fig. 6A) indicating subsetting beyond histology alone.

Reviewer #3

(Remarks to the Author)

In this paper, Heming and colleagues examine the snRNA-Seq of sural nerve samples from 33 peripheral neuropathy patients and 4 controls and compared it to spatial transcriptomics. The patients were a mix of different subtypes of chronic idiopathic axonal PN (CIAP), diabetic PN (DPN), CIDP, PPN and vasculitic neuropathy (VN). They identify several transcripts unique to human cell types. That is the only strength of the paper.

Overall, I think this is a good preliminary study. Major limitation of the study is the relatively few numbers of cases in each category, heterogeneity of the disease severity at the time of biopsy and confounders such as treatment at the time of biopsy. In data analysis there is no control for classical contributors to phenotype (age, sex, disease duration, associated conditions etc.) and attributing certain characteristics to disease etiology is premature. I think data in Figure 5 is garbage in and garbage out. They don't have enough numbers to make any conclusions, even preliminary ones. The only disease subtype that appears slightly different than others are the VN patients and even they are not completely separable from others based on this data. Therefore, conclusions such as this one in abstract "Single cell transcriptomics supported the differential diagnosis of PNP with potential for future unbiased diagnostic classification" is completely misleading and is not supported by the data (similarly comments about "personal neuropathology" elsewhere in the paper are also premature).

I think focusing on the strengths of the paper (figures 1-4) and acknowledging the severe shortcomings and eliminating any conclusions about ability to separate based on unbiased clustering, this would be a good contribution to the literature.

Version 1:

Reviewer comments:

Reviewer #1

(Remarks to the Author)

The authors have successfully addressed my concerns. I congratulate them on this study.

Reviewer #2

(Remarks to the Author)

The authors have thoughtfully addressed my concerns. The manuscript is much improved.

Reviewer #3

(Remarks to the Author)

I think the paper has significantly improved. My only comment is why do we need a new classification for lipid laden macrophages (NALAM)? It is well known that after axonal degeneration, myelin and other cellular debris is cleared by macrophages and they express the typical profile of lipid associated macrophages. I think use of the term LAM is more appropriate unless they do a direct comparison to other LAMs and demonstrate a relevant and feature set unique to nerves.
